# Composed Multicore Fiber Structure for Extended Sensor Multiplexing with Fiber Bragg Gratings

**DOI:** 10.3390/s22103837

**Published:** 2022-05-19

**Authors:** Ravil Idrisov, Adrian Lorenz, Manfred Rothhardt, Hartmut Bartelt

**Affiliations:** Leibniz Institute of Photonic Technology, 07745 Jena, Germany; ravil.idrisov@leibniz-ipht.de (R.I.); adrian.lorenz@leibniz-ipht.de (A.L.); manfred.rothhardt@leibniz-ipht.de (M.R.)

**Keywords:** multiplexing, fiber Bragg gratings, multicore fiber, photosensitivity, shape sensing

## Abstract

A novel multicore optical waveguide component based on a fiber design optimized towards selective grating inscription for multiplexed sensing applications is presented. Such a fiber design enables the increase in the optical sensor capacity as well as extending the sensing length with a single optical fiber while preserving the spatial sensing resolution. The method uses a multicore fiber with differently doped fiber cores and, therefore, enables a selective grating inscription. The concept can be applied in a draw tower inscription process for an efficient production of sensing networks. Along with the general concept, the paper discusses the specific preparation of the fiber-based sensing component and provides experimental results showing the feasibility of such a sensing system.

## 1. Introduction

Fiber gratings in optical fibers stand out in sensing solutions due, e.g., to their immunity to external electromagnetic fields, having a high sensitivity for different sensing parameters, being small in size and thereby allowing integration in many technical structures and with a high possible degree of multiplexing [1,2]. Arrays of consequently inscribed gratings with different periods enable dense wavelength division multiplexing (DWDM) in a single channel. This multiplexing method typically permit the combination of 40–80 gratings, each representing a single sensing point, in a single core fiber connected to a commercial interrogator with spectral bandwidth of about 80 nm [3,4,5]. Further increases in the grating number will usually lead to a reduced measurement range of each sensor. Due to a considerable higher attenuation in a highly photosensitive fiber core, the length of such a sensing system may also be limited. Optionally, multiple fibers can be used in parallel to extend sensing length (Figure 1a). This requires, however, complicated handling and advanced instrumentation, e.g., multichannel interrogators or fiber switches. With interrogation systems based on a scattering signal measurement, discrimination between different sensing fibers can be achieved with a single-channel device. The multiplexing principle, SLMux, was recently published and validated [6,7]. As an alternative, an array of weak gratings can be used as a sensing system using the time-of-flight measurement principle [8,9,10]. In this case, each individual grating inscribed in the fiber typically has less than 0.1% reflectivity [11] and forms a quasi-continuous grating over a length of several meters. As a consequence, the sensing length is not limited by the number of gratings inscribed but rather by the signal-to-noise ratio achieved for each of the reflected signals. Code division multiplexing is another recent development in the field of fiber Bragg gratings detection. It combines both, temporal and spectral information to identify a single sensor in a sequence [12,13,14]. This technology extends the sensing length by using sophisticated instrumentation and signal processing. A selective inscription of fiber gratings in multicore fibers is achieved by strong local laser beam focusing at a specific position inside a fiber or by limited coherence of a laser source providing an interference pattern with sufficient contrast only for part of the core area [15]. This requires very careful and stable positioning of the laser focus. It has been also demonstrated that specific fiber cores can be selected for grating inscription by proper focusing of the inscription beam [16,17,18].

In this paper, we discuss a new method to extend space-division multiplexing by exploiting a simple selective inscription concept for fiber gratings in a multicore optical fiber component. Instead of using multiple single core fibers, spatial multiplexing is achieved with a multicore optical fiber (Figure 1b). The fiber cores are differently doped such that the cores become selectively sensitive to grating inscription and, therefore, enable selective positioning of sensing points. Tailoring the photosensitivity of fiber cores has been demonstrated to be an efficient strategy for uniform FBG inscription in multicore optical fiber [19]. This makes the grating inscription simple and would also be applicable for a draw tower inscription process. In order to achieve an extended sensing length several elements of such a sensor fiber are combined to achieve an extended fiber sensor component. A general setup for such spatially divided gratings is shown in Figure 1b.

At first, we present the general concept of the fiber-sensing component more specifically. Then, the design and the fabrication of the multicore fiber as the basic elements of the sensing component are discussed. As a proof of principle, a sensor component using this concept was made consisting of two fiber elements in order to demonstrate the principle experimentally.

## 2. Materials and Methods

A sensing system based on a multicore fiber provides a more compact solution for an extended sensing range compared to a system based on multiple optical fibers. Our concept is addressing such multicore fibers combined with selective grating inscription, which avoids the handling complexity of multiple fibers (Figure 2).

We suggest achieving an inscription in a certain core by using a preform and, hence, a fiber with cores consisting of different dopants to adjust photosensitivity. Optical fibers with high germania content in the cores are known to change refractive index under UV illumination condition [20,21,22]. At the same time, many other dopant materials show no or very poor photosensitivity to UV light. Purposely incorporating both high and low photosensitive cores in a single multicore fiber achieves a selective modification of cores using a UV grating inscription process. At a specific fiber position, the complete fiber might be illuminated by the grating inscription pattern, but only in the photosensitive core would a fiber grating be produced. The other cores would guide the light to such positions without any additional losses. In this way, it would be possible to achieve sequences of local gratings with a different period for the use in combination with DWDM as well as continuous long gratings for time-of-flight-based measurements; however, for complex grating combinations it would then be required to switch the photosensitivity property along the waveguides, i.e., to change from a germania-doped waveguide property to a non-germania-doped waveguide. This is difficult to achieve in a conventional preform drawing process; therefore, we propose to provide such an effect by combining several parts of a specifically designed multicore fiber, i.e., with different rotational orientations to build up a full fiber-sensing component. As a simple example, we consider in Figure 3a a fiber cross-section with one photosensitive core and five non-photosensitive cores. By splicing six identical fiber elements under $60^{\circ }$ rotation at each splicing position we achieve a sixfold extension in the applicable sensing length. The same concept can be applied to more complex situations, e.g., for a fiber cross-section with four photosensitive and four or eight non-photosensitive cores, which have been specially adapted to shape sensing applications (Figure 3b,c). Considering an interferometric technique for fiber inscription, it would be particularly efficient to produce the fiber gratings first (e.g., during the fiber drawing process) and to split the fiber into pieces and make the recombination for the fiber-sensing component in a second step. We note that due to the use of fiber splices and due length of the gratings some restrictions have to be accepted for the possible minimum distance between grating positions.

Two preform fabrication methods can typically be used for multicore fibers with two different types of cores, rotary ultrasonic machining [23,24] and stack-and-draw [25], with the latter one being more common.

Here we describe the stack-and-draw method used for preparation of a preform for which a number of sub-preforms and filling rods are placed in a capillary tube (Figure 4a,b).

The sub-preforms are made of different single-core structures, where the core doping defines the photosensitivity. For the fiber cores with a high photosensitivity typically a high germania doping is used (15–40 mol%) [20,26]. Such concentrations elevate the refractive index of a core [26,27,28,29,30], while also being beneficial for guiding properties. The reason for this is that for multicore fibers a small mode field diameter is favorable due to reduced cross-talks between cores. On the other hand, a side effect of highly doped germania fibers is a higher transmission loss and a greater numerical aperture (NA). Increased NA might contribute to higher splicing losses when spliced to mode field mismatched fibers. To optimize splicing losses, cores of both connecting fiber ends should have a similar core size and refractive index.

The selection of the non-photosensitive sub-preform doping material should be optimized towards matching with a photosensitive core. There are a number of doping materials available with such matching properties studied for both dispersion and photosensitivity [31,32,33,34,35].

Aluminum oxide (Al_2_O_3_) is one of the dopants which provides a rather high Δn and which has been used in our case [36,37]; however, the limitation of the Al_2_O_3_ doping level of the cores is the risk of crystallization and mechanical stress after fiber drawing.

The prepared fiber preform is then then processed in a normal fiber drawing procedure (Figure 4c). The grating inscription can be achieved during fiber drawing process, before the coating application (Figure 4d). In this case, the good mechanical properties of the fiber are well preserved due to the continuous coating. The normal post-drawn procedure of fiber gratings inscription can be carried out either with a UV-transparent coating [38,39,40] or after coating removal.

The next step is cutting the fiber in such a way that each piece contains the gratings, which are supposed to be interrogated in discrete channels, rotating them in respect to each other and splicing again (Figure 4f).

As a result, the final fiber-sensing component can be achieved providing, e.g., a sixfold grating sensor capacity extension, as shown in the example in Figure 2.

## 3. Results

In order to validate the presented multicore fiber-sensing component experimentally and to show its applicability also for complex fiber structures, we were aiming for an application case for shape sensing. Shape-sensing fibers typically have multiple cores arranged in a ring for bend measurement. Outer cores are also deforming under torsion and axial strain. An additional single core in the center of the fiber always remains in the neutral axis of bending and torsion and is sensitive only to the axial strain. Thus, discrimination of axial strain is achieved by extracting equal addition of strain to all cores of the fiber. Further torsion measurement is possible by measuring equal strain change experienced by outer cores only.

To achieve similar selective sensitivity for the multicore fiber component we designed a fiber element with the outer cores arranged with equal distance from the fiber center as desirable for bending measurements (Figure 5a). The role of the central core is transferred to an off-center core gaining sensitivity to torsion. Optical fiber shape sensor performance with a similar design with a single central core was demonstrated previously [41]. In this case, axial strain discrimination is achieved via different sensitivity coefficients for axial strain and torsion applied to this core. Shifting a core from the center allows for the introduction of a “twin”-non-photosensitive core for multiplexing. Such a structure corresponds to a double-fold arrangement of a conventional shape-sensing fiber structure with four cores (Figure 5a), where the central core is shifted from the fiber center (Figure 5b,c). Additionally, we included an air channel for creating asymmetry for a simple fiber orientation control while splicing [42].

We used 15 mol% germania-doped sub-preform for the photosensitive cores providing a NA of about 0.25 and a core diameter of 4.5 μm. Sub-preforms with 5 mol% Al_2_O_3_ have been used for a preform stack. Such a low concentration implies that the numerical aperture of 0.17 is lower compared to germania-doped cores. From simulations, it is expected that the coupling efficiency in a splice of the germania-doped core and the aluminum-doped core is then limited to about 70%; however, the coupling efficiency could be further improved by an even more optimized matching of the numerical apertures.

The fiber Bragg gratings have been inscribed in a post-drawing process using a single KrF excimer laser pulse and a Talbot interferometer. Two sequences of FBGs were inscribed, each sequence having three gratings with a length of 8 mm and with 1.5 nm of spectral distance between reflection peaks. The second sequence was shifted compared to the first sequence by 1 nm towards longer wavelengths in a way that the grating reflection peaks of both sequences appear one after another in the reflection spectrum. The spectral bandwidth of the single reflection peaks was about 0.1 nm. The fiber was cut then between the sequences and cleaved normally for further splicing procedure.

As an example two fiber elements have been spliced together under a $180^{\circ }$ rotation angle as shown in Figure 5d. The accurate splicing of the pieces was performed by the Polarization Maintaining Fusion Splicer Fujikura FSM-100P+. Measured reflection peaks of two out of eight cores of the multicore fiber component as detected by a Micron Optics sm125 interrogator, are shown in Figure 5e. The result shows that all the reflection peaks are well resolved. No reflection effects from the non-photosensitive cores have been detected, which proves the feasibility of the selective grating inscription.

## 4. Discussion and Conclusions

A novel fiber-sensing component exploiting a fiber design principle that enables inscription of fiber grating arrays in separated fiber cores has been presented. The main advantage of the design is a simple realization of selective fiber grating inscription without a need for creating different illumination conditions for cores of a multicore fiber, e.g., focusing or limited coherence illumination. The selectively inscribed gratings can further be separated in several elements by cutting the fiber to single pieces and splicing them together with a rotational shift to provide transmission of the grating signal from one piece through others using unmodified cores. This allows for the creation of multiple times longer sensors compared to a single fiber element. In our experimental fiber component, we have shown a two-fold extension with a rather complex core structure; however, more fiber elements may be combined for a composed fiber and also more simple core structures, e.g., with a single photosensitive core, could present an attractive application case. Finally, we would like to note that this concept is not only applicable to fiber Bragg gratings but also to other kinds of gratings such as long period gratings.

## Figures and Tables

**Figure 1 sensors-22-03837-f001:**
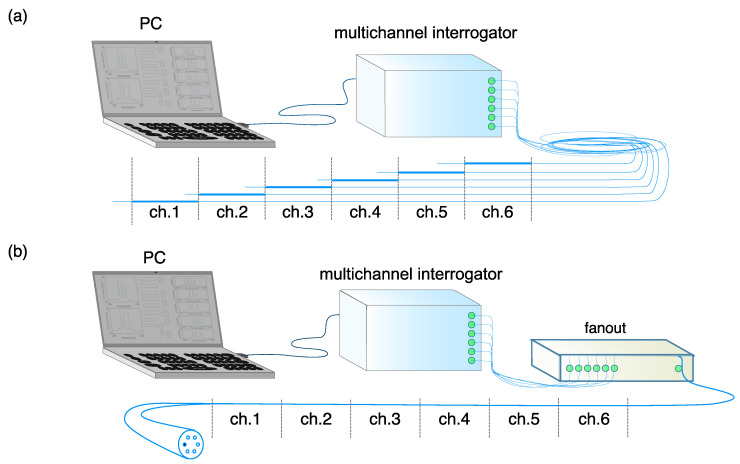
Optical schemes for distributed measurement beyond single optical fiber sensor spectral capacity: (**a**) sensing length extension with multiple single-core fibers; (**b**) sensing scheme based on a multicore fiber.

**Figure 2 sensors-22-03837-f002:**
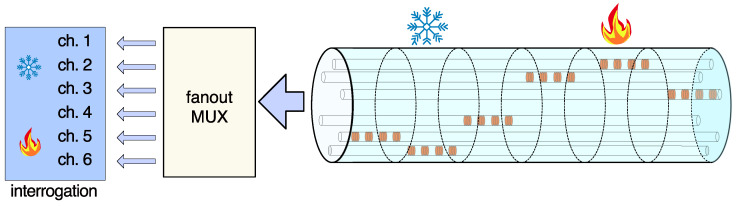
Optical fiber sensing networks with extended length based on a multicore fiber with selectively inscribed FBGs in the cores. In case of temperature measurements performed in both cases, the local temperature changes are detected in discriminated channels.

**Figure 3 sensors-22-03837-f003:**
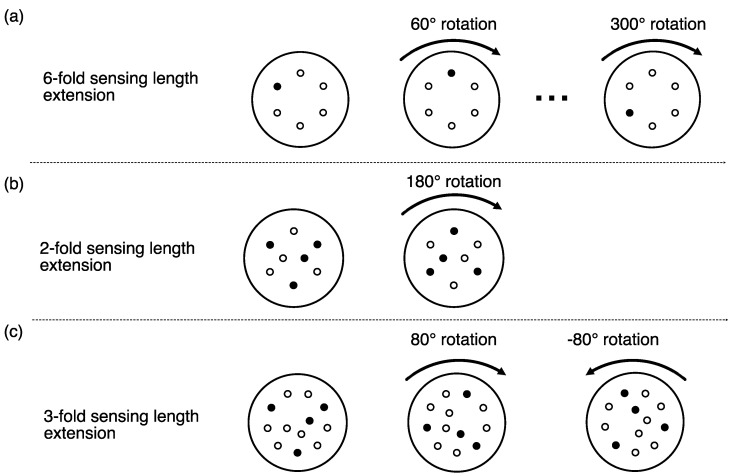
Fiber designs examples which can be used for a multicore fiber component. (**a**) Sixfold length extension optical fiber design with a single sensing core. (**b**) Two- and three-fold (**c**) length extension design for shape sensing fiber with four sensing cores.

**Figure 4 sensors-22-03837-f004:**
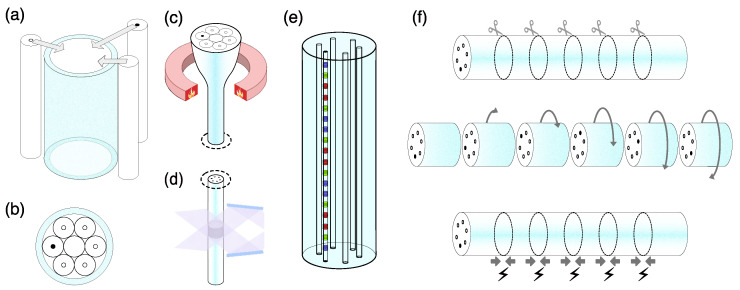
Step-by-step process of a concept fiber component manufacturing. (**a**) Preform stacking. The core material is made from a few sub-preforms with tailored photosensitivities, space between sub-preforms is filled with silica rods. (**b**) Cross-section of a stacked preform. (**c**) Fiber drawing process. (**d**) Fiber Bragg gratings can be inscribed during fiber drawing process. (**e**) The resulting fiber has FBGs inscribed only in photosensitive cores. (**f**) Further fiber handling and component preparation: the fiber is cleaved to obtain a fiber with arrays of FBGs. Then different parts are connected and spliced.

**Figure 5 sensors-22-03837-f005:**
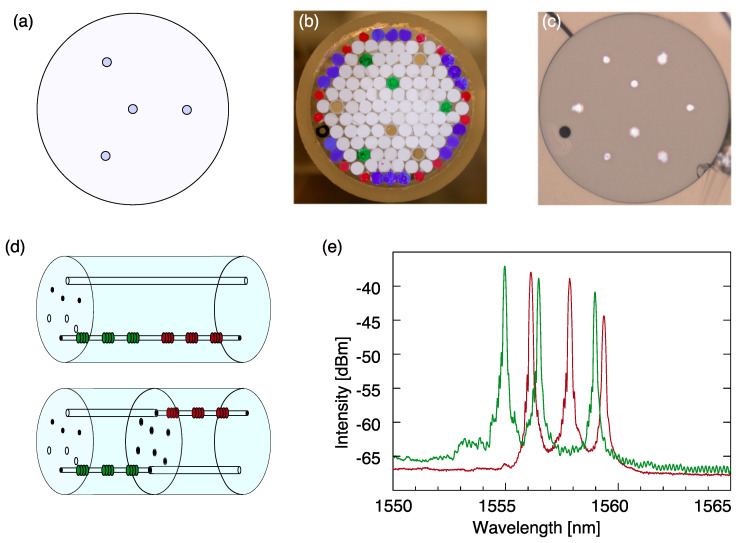
Experimental proof-of-concept of the multiplexing fiber component. (**a**) Cross-section of a typical shape sensing fiber. (**b**) Stacked preform for the multiplexing fiber element with photosensitive sub-preforms marked green, insensitive sub-preforms in dark-beige, filling rods of two different size are marked with violet and red. (**c**) Microscope image of the drawn fiber, larger diameter circles correspond to the cores insensitive to UV illumination, small diameter circles represent the photosensitive cores. A small capillary (black) was introduced in the preform for assuring the required orientation. (**d**) Fiber component is prepared by splicing two pieces with a rotational shift of $180^{\circ }$. (**e**) Spectra of FBGs in two cores out of eight (green spectrum corresponds to the FBGs depicted in green in (**d**), and red spectrum correspond to the FBGs depicted in red in (**d**)).

## Data Availability

Data is available upon request. Please contact the corresponding author.

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
