# Peer review of "Composed Multicore Fiber Structure for Extended Sensor Multiplexing with Fiber Bragg Gratings"

_sensors, 2022, doi:10.3390/s22103837_

Round 1
Reviewer 1 Report
THe authors present a multiplexing scheme for multicore fibers and its possible application. Overall, the work is acceptable but requires a revision in two main parts:
1) The main alternative to the scheme proposed by the authors is the scattering-level multiplexing (SLMux)that has been recently presented for fibers with enhanced backscattering. In the introduction, or in a discussion, the authors should compare their novel method with SLMux
2) A simple detection experiment should be included, showing the possibility to multiplex data (for example, a temperature or strain experiment) to better validate the concept
Author Response
Authors appreciate the valuable comments and thorough text revision provided by the reviewer.
1. Indeed, the SLMux is an interesting multiplexing method that provides a possibility to extend a sensing length limitation of conventional scattering signal-based sensors. We added a note on the method:
‘It is worth mentioning that with interrogation systems based on scattering signal measurement, a discrimination between different sensing fibers can be achieved with a single-channel device. The multiplexing principle, SLMux, was recently published and validated [ 6,7 ].’
2. We understand the interest of the Reviewer on some results as an FBG sensor. However, the main focus of the article was to present the concept and a proofe of the feasibility of the multiplexing method, as the sensing mechanism remains the same and is expected to perform in a similar way for a fiber with solely photosensitive cores and with FBGs inscribed in the cores. For the multiplexing functionality, the experimental results of fig. 5 have been described. In order to mention experimental results for sensing we added a note about using a fiber with comparable design (7 core instead of 8 core with similar core-to-center distance) as a shape sensor which can be found in the reference [41].
Reviewer 2 Report
In this article, the authors present a multicore fiber component concept to improve multiplexing capability. I have some concerns about the proposal:
- Throughout the manuscript, the authors discuss the sensor's potential for applications in shape and curvature sensing; they also present an interesting configuration for these applications in figure 4f. However, such applications and structure are not demonstrated in the experimental results. The authors should give experimental results that confirm the possibility of such applications, for example, showing the device response in curvature tests.
- Authors need to give more experimental details, such as how the device was interrogated (source, detector, etc.), how rotation and splicing were performed (splicing machine, parameters, how the cleavage position was determined, etc.).
- In section 3, page 5, lines 158 and 159, the authors state that "No reflection effects from the non-photosensitive colors have been detected...", however in figure 5e, it is possible to notice an interference pattern in the green spectrum that could be the formation of a cavity due to reflection at the splice between the cores.
- Regarding the manuscript form, a language review and a conclusion section are necessary.
If these concerns are clarified, and new application results are presented, the manuscript may have merit for publication.
Author Response
Authors appreciate the valuable comments and thorough text revision provided by the reviewer.
- We understand the interest of the Reviewer on some results as an FBG sensor. However, the main focus of the article was to present a concept and prove the feasibility of the presented new multiplexing method, as the sensing mechanism remains the same and is expected to perform similar to a fiber with solely photosensitive cores with FBGs inscribed in the cores. Detailed experimental results on the use of multicore fibers for shape sensing have been published in other papers, e.g. in reference [41]. In order to mention such experimental results we added a note about using a fiber with comparable design (7 core instead of 8 core with similar core-to-center distance) as a shape sensor which can be found in the reference [41].
- The following details were added to give more experimental data for the used systems (standard parameters have been applied):
‘The accurate splicing of the fiber pieces was performed by the Polarization Maintaining Fusion Splicer Fujikura FSM-100P+.
The measured reflection peaks of two out of eight cores of the multicore fiber component, as detected by a Micron Optics sm125 interrogator, are shown in Fig. 5e’
- The reason for the small artefact in the green spectrum has not been investigated in detail. With our text we wanted to indicate that no reflections have been detected from possible weak FBGs in the non-photosensitive cores. The text has been modified accordingly:
“No reflection effects from the non-photosensitive cores have been detected, which proves the feasibility of the selective grating inscription.”
4. The text has been checked by a native speaker for English language quality.
We would like to point out that the “Discussion” section presents also conclusion aspects of the presented results. Since a conclusion section is not mandatory, we wanted to avoid repeating the arguments already mentioned and discussed before in another “Conclusions” section. To make this point clearer we have renamed the section: “Discussion and Conclusions”
Reviewer 3 Report
This paper reports a composed multicore fiber structure for extended sensor multiplexing with fiber gratings. I have some comments.
- Title needs to be changed since the principle is shown only for FBG and not for all fiber gratings. although it is mentioned in the discussion "This concept is not only applicable to fiber Bragg gratings but also to other kinds of gratings such as long period gratings." it is not shown and there is not data. Please comment.
- In additional, why refer multiplexing sensor if not explored in detail it?
- Introduction needs improvement about the importance of first sentence written. Include some literature about these advantages for physical and biochemical sensing probes: Optics Express 30 (10), 16518-16529, 2022; Optics & Laser Technology 140, 107082, 2021.
- How about the reproducibility to get similar FBGs performance?
- How about the create a multiparameter? Please explore.
Author Response
Authors appreciate the valuable comments and thorough text revision provided by the reviewer.
- The title has been modified according to the reviewers suggestion: ‘Composed multicore fiber structure for extended sensor multiplexing with fiber Bragg gratings’.
- It was the intention of this paper to propose and to present a proof-of-concept of the new multiplexing concept. For this purpose, the experimental results of fig. 5 have been described. Results for the specific application case of shape sensing with multicore fibers have been presented in other papers. In order to mention such experimental results we added a note about using a fiber with comparable design (7 core instead of 8 core with similar core-to-center distance) as a shape sensor which can be found in the reference [41].
- The reference list has been updated accordingly [1,2].
- The reproducibility of the inscribed FBGs depends on a few factors: optical fiber uniformity and inscription conditions stability. The difference that can be seen on the FBGs spectra is explained by the excimer laser pulse energy fluctuation. With stable illumination conditions we expect high FBGs repeatability.
- Indeed, multi-parameter sensing with fiber gratings has been discussed and demonstrated in the literature. Specifically, the use of different materials might enable different sensitivity to strain, temperature and other parameters. However, the main focus of this manuscript is related to the use of different doping materials in the cores for space division multiplexing and not for multi-parameter sensing.
Round 2
Reviewer 1 Report
My comments have been addressed
Reviewer 2 Report
The authors have clarified concerns; the manuscript is suitable for publication.
Reviewer 3 Report
I am happy with the improvements done.